# RNF8 Dysregulation and Down-regulation During HTLV-1 Infection Promote Genomic Instability in Adult T-Cell Leukemia

Huijun Zhi[1☯], Xin Guo[2☯], Yik-Khuan Ho[1☯], Nagesh Pasupala[1], Hampus Alexander Anders Engstrom[2], Oliver John Semmes[2]*, Chou-Zen Giam[1]*

**1** Department of Microbiology and Immunology Uniformed Services University of the Health Sciences Bethesda, MD, United States of America, **2** Department of Microbiology and Molecular Cell Biology The Leroy T. Canoles Jr Cancer Research Center Eastern Virginia Medical School Norfolk, VA, United States of America

☯ These authors contributed equally to this work.
* semmesoj@evms.edu (OJS); chou-zen.giam@usuhs.edu (C-ZG)

**Data Availability Statement:** All relevant data are within the manuscript and its Supporting Information files.

## Abstract

The genomic instability associated with adult T cell leukemia/lymphoma (ATL) is causally linked to Tax, the HTLV-1 viral oncoprotein, but the underlying mechanism is not fully understood. We have previously shown that Tax hijacks and aberrantly activates ring finger protein 8 (RNF8) — a lysine 63 (K63)-specific ubiquitin E3 ligase critical for DNA double-strand break (DSB) repair signaling — to assemble K63-linked polyubiquitin chains (K63-pUbs) in the cytosol. Tax and the cytosolic K63-pUbs, in turn, initiate additional recruitment of linear ubiquitin assembly complex (LUBAC) to produce hybrid K63-M1 pUbs, which trigger a kinase cascade that leads to canonical IKK:NF-κB activation. Here we demonstrate that HTLV-1-infected cells are impaired in DNA damage response (DDR). This impairment correlates with the induction of microscopically visible nuclear speckles by Tax known as the Tax-speckle structures (TSS), which act as pseudo DNA damage signaling scaffolds that sequester DDR factors such as BRCA1, DNA-PK, and MDC1. We show that TSS co-localize with Tax, RNF8 and K63-pUbs, and their formation depends on RNF8. Tax mutants defective or attenuated in inducing K63-pUb assembly are deficient or tempered in TSS induction and DDR impairment. Finally, our results indicate that loss of RNF8 expression reduces HTLV-1 viral gene expression and frequently occurs in ATL cells. Thus, during HTLV-1 infection, Tax activates RNF8 to assemble nuclear K63-pUbs that sequester DDR factors in Tax speckles, disrupting DDR signaling and DSB repair. Down-regulation of RNF8 expression is positively selected during infection and progression to disease, and further exacerbates the genomic instability of ATL.

## Author summary

Approximately 3–5% of HTLV-1-infected individuals develop an intractable malignancy called adult T cell leukemia/lymphoma (ATL) decades after infection. Unlike other

**Funding:** This work was supported by grants from the National Institutes of Health (R21CA216660) and Uniformed Services University (HU0001-14-1-0061) to C.-Z. G. The funders had no role in study design, data collection and analysis, decision to publish, or preparation of the manuscript.

**Competing interests:** The authors have declared that no competing interests exist.

leukemia, ATL is characterized by extensive genomic instability. Here we show that the genomic instability of ATL is associated with the hijacking and aberrant activation of a molecule known as ring finger protein 8 (RNF8) by HTLV-1 for viral replication. RNF8 is crucial for initiating the cellular DNA damage response (DDR) required for the repair of DNA double-strand breaks (DSBs), the most deleterious DNA damage. Its dysregulation in HTLV-1-infected cells results in the formation of pseudo DNA damage signaling scaffolds known as Tax speckle structures that sequester critical repair factors, causing an inability to repair DSBs efficiently. We have further found that loss of RNF8 expression reduces HTLV-1 viral replication and frequently occurs in ATL of all types. This likely facilitates the immune evasion of virus-infected cells, but degrades their ability to repair DSBs and exacerbates the genomic instability of ATL cells. Since DDR defects impact cancer response to DNA-damaging radiation and chemotherapies, RNF8 deficiency in ATL may be exploited for disease treatment.

## Introduction

HTLV-1 is a complex human delta retrovirus that currently infects 10–20 million people worldwide [1]. While HTLV-1 infection is generally asymptomatic, 3–5% of infected individuals develop an intractable T cell neoplasm known as adult T cell leukemia/lymphoma (ATL) decades after infection [2, 3]. How HTLV-1 infection progresses to ATL is not fully understood.

Unlike cells of other hematological malignancies [4, 5], ATL cells often contain complex genomic and chromosomal abnormalities including base substitutions, insertions and deletions (indels), gene rearrangement and amplification, and aneuploidy [6]. In agreement with these earlier findings, a recent whole-genome/exome/transcriptome analysis of ATL has revealed an average of 2.3 point mutations/Mb and 59.5 structural variations per ATL genome [7], almost 3 times as frequent as in multiple myeloma (21 structural variations/genome) [4].

Dysfunction in DNA damage response (DDR) is associated with genomic instability and predisposition to cancer. It also impacts how malignant cells respond to genotoxic chemotherapies [8]. The cause for the genomic instability of ATL has been attributed to the HTLV-1 trans-activator/oncoprotein, Tax, which, in addition to being a potent activator of viral transcription and I-κB kinase (IKK)/NF-κB signaling, is a clastogen that promotes frequent DNA double-strand breaks (DSBs), resulting in micronuclei formation [9–12]. Tax not only actively causes DSBs, but also represses DDR. In the presence of Tax, DDR induced by ionizing radiation is inhibited, leading to persistent DNA damage and DDR signaling [13]. We (Semmes et al.) have previously shown that Tax induces the formation of nuclear foci termed Tax-speckle structures (TSS) that contain Tax, BRCA1, DNA-PK, and MDC1, but not Nbs1 [14]. While TSS sequester DNA damage repair factors, how Tax promotes their formation is not known.

Using cell-free systems reconstituted with purified proteins and cell-based assays, we have shown previously that Tax hijacks and aberrantly activates a ubiquitin E3 ligase, ring finger protein 8 (RNF8), and its associated E2 conjugating enzymes, Ubc13:Uev1a/Uev2 (Ubc13 for brief), to assemble cytosolic lysine 63-linked polyubiquitin (K63-pUb) chains for TAK1, IKK/NF-κB, and JNK activation [15]. RNF8 is a major mediator of DDR in the nucleus. It is recruited to the sites of DSBs to covalently attach K63-pUb chains to histones H1, H2A, and H2AX [16–18], initiating the formation of signaling scaffolds where a large assembly of DDR factors congregate to form microscopically visible DNA damage foci.

Here we show that HTLV-1-infected cells are impaired in DNA damage response (DDR). This impairment appears to involve a functional sequestration of activated RNF8 by Tax and correlates with the induction of TSS. We demonstrate that TSS contain both Tax and K63-pUb chains, and their formation requires RNF8 whose ablation results in the disappearance of TSS from the nuclei of Tax-expressing cells. Analyses of two Tax mutants, M22 and M47, indicate that robust RNF8 activation and K63-pUb chain assembly are important for TSS induction and DDR repression. We have further found that loss of RNF8 reduces HTLV-1 gene expression, and many ATL cell lines express RNF8 mRNA and protein at low to undetectable levels. *In silico* analysis of mRNA microarray data of ATL patients further revealed frequent RNF8 down-regulation in ATL of all types. Thus, Tax hijacks RNF8 to assemble K63-pUbs in both cytosolic and nuclear compartments. The cytosolic K63-pUbs initiate a kinase cascade that leads to IKK/NF-κB activation, while the nuclear K63-pUbs sequester critical DDR factors into TSS, disrupting DDR. Loss of RNF8 mitigates NF-κB activation by Tax, reduces viral gene expression, and is positively selected during ATL development. RNF8 deficiency, in turn, further exacerbates the genomic instability of ATL.

## Results

### HTLV-1-infected cells are defective in DNA damage response (DDR)

The molecular basis for the genomic instability of ATL is incompletely understood. Previous studies have indicated that it is causally linked to the viral trans-activator/oncoprotein, Tax [9–12]. Tax not only actively causes DNA damage [10, 11], but also represses DNA repair [13]. Earlier experiments demonstrating the impact of Tax on genomic instability were performed under conditions where Tax is over-expressed after DNA transfection. Whether physiological levels of Tax produced during viral infection have the same effect has not been examined. The study has also been hampered by the fact that most *tax*-transduced or HTLV-1-infected cells become senescent or cell-cycle arrested [19]. We have shown previously that the Tax-induced rapid senescence (Tax-IRS) is driven by NF-κB hyperactivation. In cells where NF-κB activity is blocked by a degradation-resistant form of IκBα, ΔN-IκBα, Tax-IRS is prevented [20]. Cell lines chronically and productively infected by HTLV-1 can therefore be established in the cellular background where NF-κB is inhibited by ΔN-IκBα [19–21].

To determine the impact of HTLV-1 infection on DDR, we used a pair of "isogenic" cell lines, HeLa-G: ΔN-IκBα and its HTLV-1-infected progeny, HeLa-G: ΔN-IκBα:HTLV-1. Cells of both lines were grown in culture, treated with 12 μM bleomycin for 1 hr to induce DNA damage, and then harvested at 0, 1, 3, 8, and 16 hrs post-treatment, and monitored for DNA damage signaling by γH2AX immunoblotting. In uninfected HeLa-G: ΔN-IκBα cells, the levels of γH2AX rose within the first 3 hours after bleomycin-induced DNA damage, but subsided at around 8 hrs post-treatment presumably when DNA damage became repaired (Fig 1A odd-number lanes). In contrast, in HTLV-1-infected cells (even-number lanes, p19-Gag-positive), γH2AX levels rose to a greater level with similar kinetics after bleomycin treatment, but persisted at higher levels over most of the 16-hr time course, suggesting an inability to repair damage. Notably, in the absence of bleomycin treatment, a low level of γH2AX was also observed in HeLa-G: ΔN-IκBα:HTLV-1, but not in uninfected HeLa-G: ΔN-IκBα cells (compare Fig 1A lanes 2 & 1), suggesting that the basal cellular DNA damage could not be properly repaired in HTLV-1-infected cells. These results lend support to the notion that DDR is impaired in HTLV-1-infected cells, likely via the action of Tax.

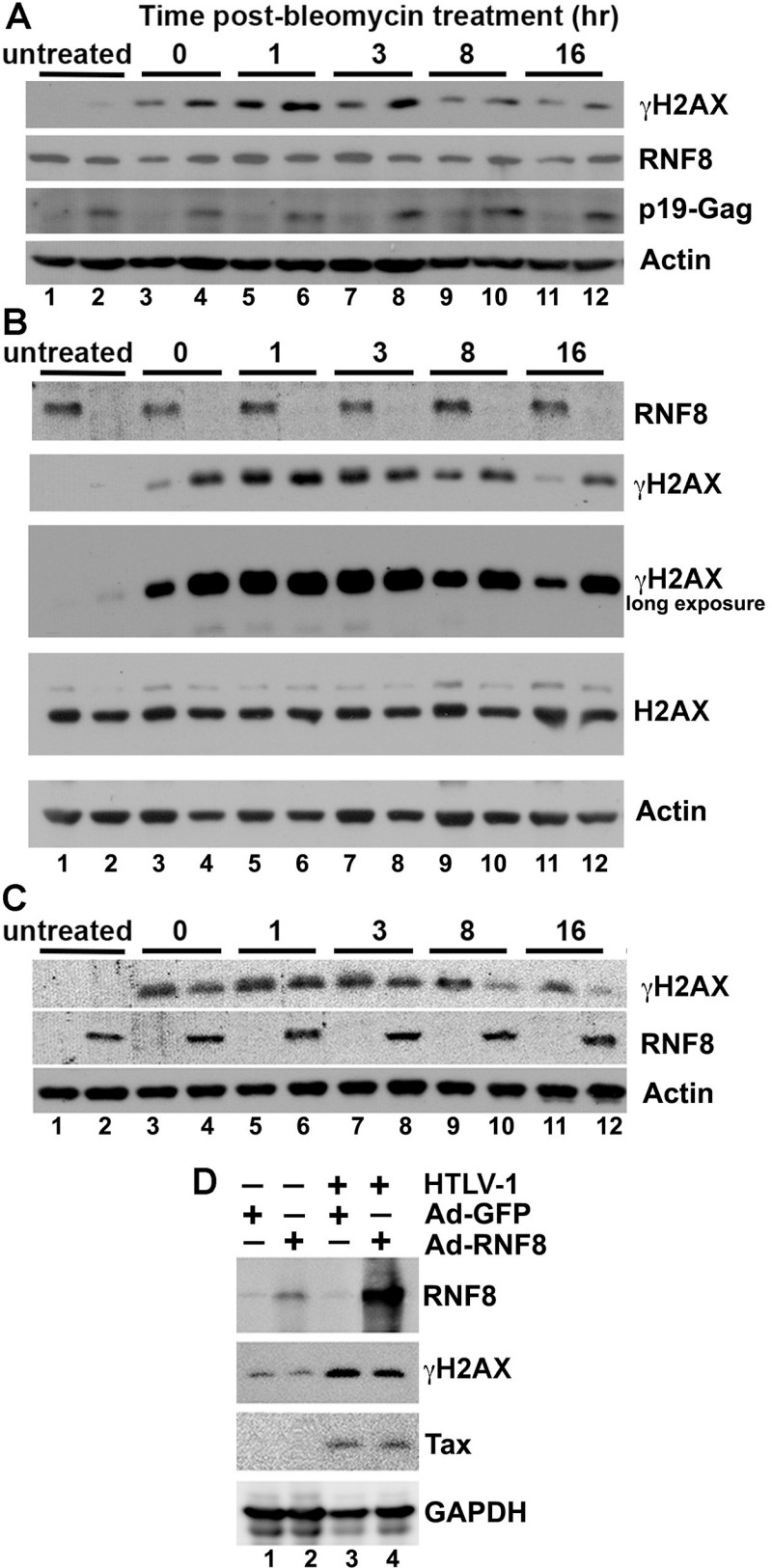

**Fig 1. The DDR impairment in HTLV-1-infected HeLa cells cannot be rescued by RNF8 over-expression. (A)** HTLV-1-infected HeLa cells are DDR impaired. HeLa-G: ΔN-IκBα and its progeny, HeLa-G: ΔN-IκBα:HTLV-1, that

has been stably infected by HTLV-1 were treated with 12 μM bleomycin for 1 hr to induce DNA damage, washed, grown in fresh bleomycin-free media and harvested at the indicated times post-treatment (0, 1, 3, 8 and 16 hrs) and monitored by IB for γH2AX, RNF8, p19 matrix protein, and β-actin control. **(B)** RNF8 is crucial for the repair of DNA double-strand breaks. HeLa-G and HeLa-G$^{\Delta RNF8}$ were similarly treated with bleomycin as in (A) and monitored by IB for RNF8, γH2AX, H2AX and β-actin control. **(C)** RNF8 over-expression rescued the DDR defect in RNF8-null cells. HeLa-G$^{\Delta RNF8}$ cells (2X10$^5$) grown in 6-well plates were infected by Ad-GFP (odd-number lanes) or Ad-RNF8 (even-number lanes) at an MOI of 5 for 16 hrs. The infected cells were treated with bleomycin and monitored for rescue at indicated times by γH2AX, RNF8, and β-actin IB. **(D)** RNF8 over-expression could not rescue the DDR defect of HTLV-1-infected cells. HeLa-G: ΔN-IκBα (lanes 1 & 2) and HeLa-G: ΔN-IκBα:HTLV-1 (lanes 3 & 4) were infected by Ad-GFP or Ad-RNF8 at MOI of 5, respectively. Twenty-four hours post-infection, cells were treated with bleomycin as in (A), and grown in bleomycin-free media for another 16 hrs. Thereupon, cells were harvested and immunoblotted for the RNF8, γH2AX, Tax and GAPDH. See Supplemental Information for quantitation of representative immunoblots.

## DDR defect in HTLV-1-infected cells cannot be rescued by RNF8 over-expression

Using cell-based assays and cell-free systems reconstituted from purified proteins, we have previously demonstrated that Tax interacts with and aberrantly stimulates RNF8 and Ubc13 to assemble long K63-pUbs *in vivo* and *in vitro* [15]. Our results together with those described by Shibata et al. [22] indicate that Tax hijacks and aberrantly activates RNF8 to assemble K63-pUbs, the Tax-RNF8-K63-pUbs complex then additionally enlist the linear ubiquitin (M1-pUb) assembly complex (LUBAC) to produce hybrid K63- and M1-pUbs that form the signaling scaffolds for the recruitment and activation of TAK1, IKK:NF-κB, JNK, and a plethora of other Ser/Thr kinases.

RNF8 is crucial for DDR signaling and DNA damage repair. Mice with homozygous deletion of the RNF8 gene, while viable, are impaired in immunoglobulin heavy chain class switching and spermatogenesis, and are highly sensitive to ionizing radiation and predisposed to tumorigenesis [23, 24]. In response to DSBs, ataxia telangiectasia mutated kinase (ATM) becomes recruited to the site of DNA damage where it phosphorylates H2AX that accumulates near DSBs. Phospho-H2AX (γH2AX) then recruits MDC1 (mediator of DNA damage checkpoint 1) to the site of DNA damage to be phosphorylated by ATM. RNF8, in turn, binds p-MDC1 via its NH$_2$-terminal FHA domain, becomes activated and covalently attaches K63-pUb to linker histone H1 [18, 25, 26]. This leads to the additional recruitment of RNF168, a K63-pUb-binding E3 ligase that amplifies and propagates DDR signaling by linking K63-pUb to histone H2A at DSBs [16, 17, 25, 27–29].

In light of the importance of RNF8 in DNA damage repair, we reasoned that Tax could repress DNA repair by sequestering or mislocalizing RNF8 to cause a deficiency in RNF8 function. Through subsequent activation of RNF8, Tax could also induce the formation of nuclear K63-pUbs clusters that sequester and mislocalize DDR factors that are normally targeted to sites of DSBs for DNA repair. To test the first possibility, we subjected HeLa-G and its RNF8-null counterpart, HeLa-GΔ$^{RNF8}$, to bleomycin treatment. In agreement with the importance of RNF8 in DSB repair, the loss of RNF8 caused the γH2AX signal induced by bleomycin to rise and persist in a manner similar to HTLV-1 infection, (Fig 1B compare odd [wildtype] and even [ΔRNF8] lanes). Untreated HeLa-GΔ$^{RNF8}$ cells, like their HTLV-1-infected counterparts, expressed a low but detectable level of γH2AX (Fig 1B lane 2 long exposure), indicating that in the absence of RNF8, DNA damage that arose spontaneously persisted, resulting in γH2AX accumulation. As expected, the DDR defect in HeLa-GΔ$^{RNF8}$ cells was rescued by restoring RNF8 expression using Ad-RNF8, a replication-defective adenovirus expression vector for RNF8 (Fig 1C compare odd- and even-numbered lanes).

We next tested whether RNF8 over-expression could correct the DDR defect caused by HTLV-1 infection. To this end, RNF8 was over-expressed in HeLa-G: ΔN-IκBα and HeLa-G: ΔN-IκBα:HTLV-1 cells by Ad-RNF8 transduction (Fig 1D lanes 2 & 4). The same cells were also transduced with Ad-GFP as a control (lanes 1 & 3). Twenty four hours after transduction, DNA damage was induced by bleomycin as before, and γH2AX levels monitored at 16 hrs post-treatment. In HeLa-G: ΔN-IκBα, γH2AX levels subsided at 8 and 16 hrs post-treatment when DNA damage is repaired (lanes 1 & 2, also see Fig 1A & 1B lanes 11 for comparison). As anticipated, RNF8 over-expression in HeLa-G: ΔN-IκBα had little impact as endogenous RNF8 was sufficient for the repair (lane 2). If the DDR defect in HTLV-1-induced cells were due to a competitive repression of RNF8, then the level of bleomycin-induced γH2AX should have decreased upon RNF8 over-expression. This, however, was not observed (lane 4). Rather, increased levels of γH2AX persisted in HTLV-1-infected cells irrespective of RNF8 over-expression (compared lanes 3 and 4). It should be pointed out that RNF8 expression in Ad-RNF8-transduced HTLV-1-infected cells was exceedingly high due to Tax trans-activation. The immunoblot was therefore exposed only briefly so as not to blackened the film. As a result, the endogenous RNF8 was minimally detectable. Based on these results, we concluded that RNF8 deficiency is not responsible for the DDR impairment of HTLV-1-infected cells.

## Formation of Tax speckle structures requires RNF8

We (Semmes lab) have previously reported that Tax induces the formation of microscopically visible and DNA damage-independent nuclear foci known as Tax speckle structures (**TSS**). TSS contain DDR mediators including BRCA1, CHK2, DNA-PKcs, and MDC1, but not Nbs1 [13, 14, 30, 31]. As RNF8 assembles K63-pUbs for DDR signaling, we ask whether the nuclear K63-pUbs assembled by aberrantly activated RNF8 may be responsible for TSS formation. To this end, we derived a RNF8-null U2OS cell line, U2OS-ΔRNF8, via CRISPR-Cas9 (Fig 2C). Wildtype and U2OS-ΔRNF8 were then transfected with Tax and visualized for TSS and K63-pUbs by immunofluorescence. In wildtype U2OS cells, TSS were readily detected (Fig 2A, U2OS, lower left panel). They co-localized with K63-pUbs signals in the nucleus (Fig 2A lower right panel). In contrast, in U2OS-ΔRNF8, Tax localized primarily to the cytoplasm, and no or few nuclear TSS signals were detectable (Fig 2B, U2OS-ΔRNF8). This conclusion was further supported by determining and comparing the number of Tax speckles per cell in 50 U2OS and U2OS-ΔRNF8 cells each (Fig 2D). As expected, RNF8 was prominently featured in TSS (Fig 2E); and nuclear TSS that contained colocalized Tax and K63-pUbs could be easily seen in HeLa cells as well (Fig 2F). These results indicate that TSS induction requires RNF8, and TSS most likely originate from the nuclear K63-pUbs assembled by RNF8 upon Tax activation, and act as mislocalized signaling scaffolds where a large assembly of misdirected DDR factors congregate. Interestingly, while cytosolic Tax-RNF8-K63-pUbs complex recruits LUBAC to produce hybrid K63-M1 pUbs, no linear M1-pUb chains were detected in TSS (not shown).

## Tax mutants impaired in TSS formation do not repress DDR

Because canonical NF-κB activation by Tax requires RNF8 [15], we examined two well characterized Tax mutants: M22 (T130A L131S) and M47 (L319R L320S) that are defective in NF-κB and LTR activation respectively [32] for RNF8 activation and TSS induction. HeLa cells were co-transfected with HA-Ub and wildtype (WT), M47, and M22 *tax* alleles for 48 hrs. Cell lysates were immunoblotted for HA (to detect pUbs), Tax, IκBα, and β-actin control (Fig 3A). As expected, M22's defect in NF-κB activation correlated with its inability to stimulate pUbs assembly and induce IκBα degradation (Fig 3A lane 4). While M47 induced IκBα degradation (Fig 3A lane 3), its ability to stimulate pUbs assembly was moderately reduced compared to

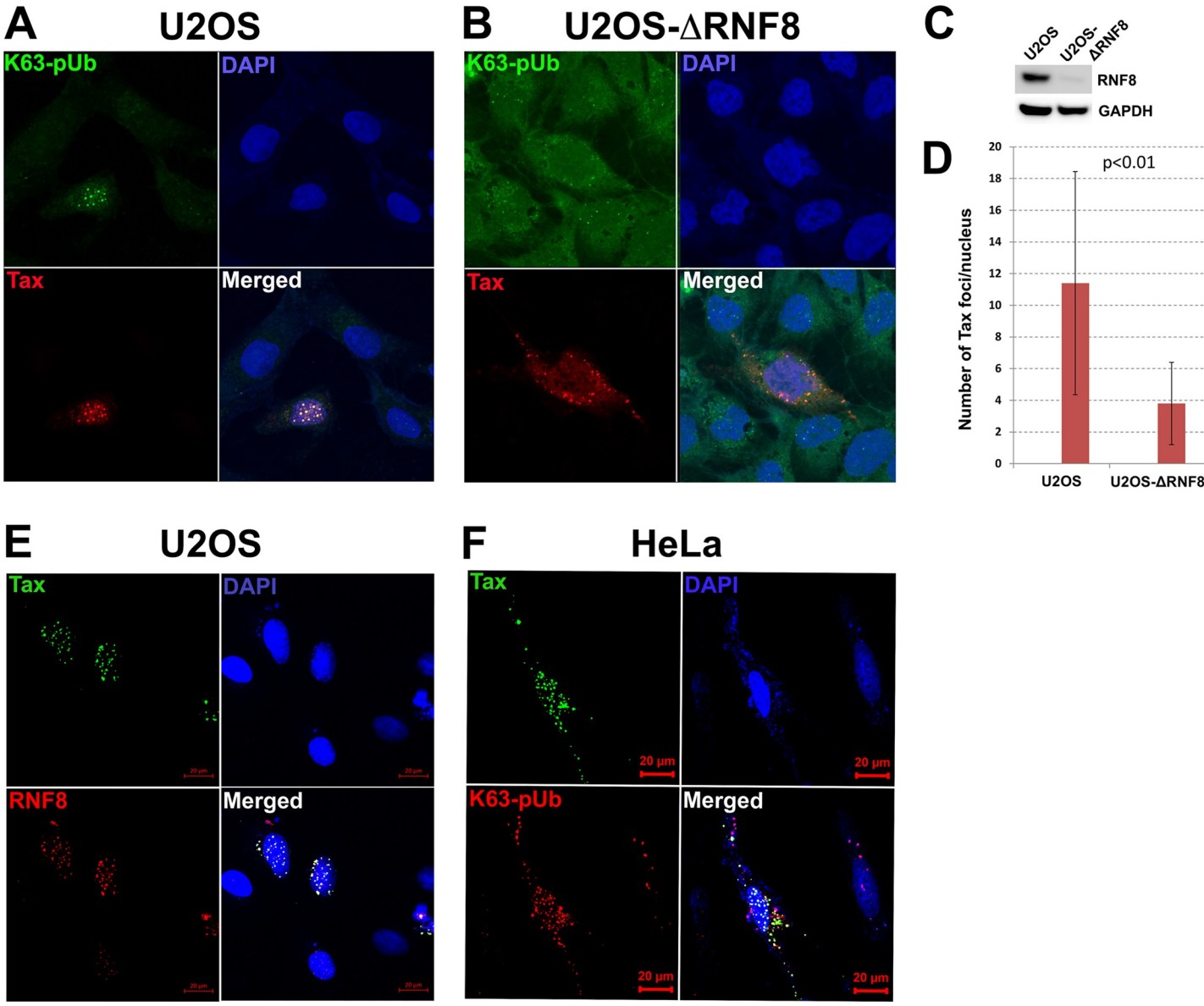

**Fig 2. Formation of nuclear Tax speckle structures (TSS) requires RNF8. (A)** U2OS and **(B)** U2OS ablated for RNF8 (U2OS-ΔRNF8) were transfected with a Tax plasmid for 48 hrs, respectively. Immuno-fluorescence was then performed to detect Tax (red) in TSS and K63-pUb chains (green, K63-TUBE (fluorescein)). Co-localization of TSS and K63-pUb chains in the nucleus (blue) are shown in the merged images. **(C)** Immunoblots of RNF8 and GAPDH expressed in U2OS and U2OS-ΔRNF8 cells. **(D)** The average numbers of TSS (Tax foci) in 17 *tax*-transfected U2OS and U2OS-ΔRNF8 cells were counted and compared using the student t-test. **(E)** U2OS and **(F)** HeLa cells were transfected with an expression plasmid encoding the S-tagged Tax-GFP fusion for 48 hrs as in **(A)**. Fluorescence microscopy was performed to detect **(E)**Tax (green) and RNF8 (red, by immunofluorescence) or **(F)** Tax and K63-linked polyubiquitin chains (K63-TUBE (TAMRA)) in TSS in the nucleus (blue). Co-localization of TSS and RNF8 or TSS and K63-pUb are shown in the respective merged images.

WT (Fig 3A compare lane 3 and 2, also see S4 Fig for quantitation). We next transfected HEK293T cells with expression constructs for S-peptide-tagged WT, M47, and M22 Tax, and used RNase S-conjugated agarose beads to capture Tax molecules and associated proteins [15]. As shown in Fig 3B, RNF8 interacted with WT and M47, but not with M22 Tax. Thus, the abilities of the respective Tax molecules to stimulate pUb chains assembly correlated with their binding to RNF8 (Fig 3B). Fluorescence imaging (Fig 3C) and a quantitative analysis of TSS (Fig 3D) further indicated M22 to be severely impaired, while M47, minimally deficient in TSS

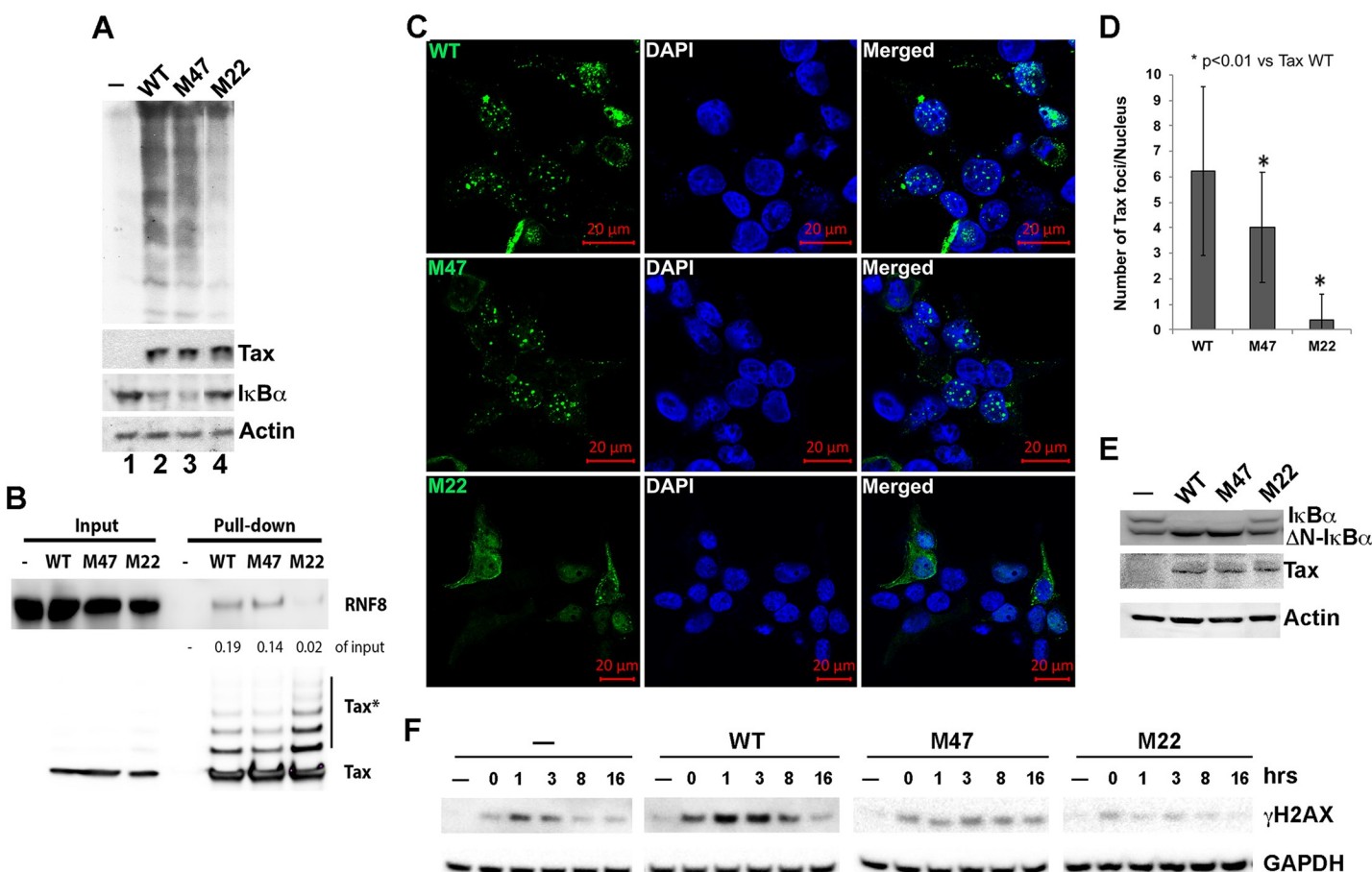

**Fig 3. Aberrant RNF8 activation by Tax causes DDR impairment.** (A) HeLa cells ($10^5$) were co-transfected for 48 hrs with HA-Ub and an expression plasmid for each of the *tax* alleles: 50 ng each for wildtype (WT) and M47, and 100 ng for M22, using lipofection (FugeneHD, Promega). Cells were harvested, lysed, and immunoblotted for HA, Tax, IκBα and β-actin control (see Supplemental Information S4 Fig for quantitation of the polyubiquitin signals). (B) Derivation of the S-tagged Tax (WT) expression construct and S-tag pull-down using the RNase S-agarose beads had been previously described [15]. Constructs for Tax mutants: M47 and M22 were similarly derived. HEK 293T cells were transiently transfected with WT, M47 and M22 Tax alleles. Tax proteins in the respective whole cell lysates were enriched using the RNase S-agarose beads (Novagen) and analyzed by Western blotting using RNF8 and Tax antibodies. Pull-down RNF8 bands were quantified using Image J and normalized to the input RNF8 (left panel). Tax* denotes post-translationally modified forms Tax variants. The S-tagged Tax was released from the beads using a large excess of S peptide and immunoblotted for Tax and RNF8 as indicated. (C) Immunofluorescence of Tax (green), nuclei (blue, DAPI), and merged images were performed as in Fig 2 at 48 hrs post-transfection of the respective *tax* alleles (upper left corners of panels in the left column) into 293 cells. (D) The number of Tax foci (i.e.,TSS) per nucleus induced by WT, M47 and M22 alleles were calculated after counting nuclear speckles of 56 transfected cells each. (E) Immunoblots (IκBαTax, and β-actin) of HeLa-G: ΔN-IκBα stably expressing WT, M47 and M22 *tax* alleles respectively. (F) HeLa-G: ΔN-IκBα:WT, M47 and M22 were treated with bleomycin as in Fig 1. DDR repression caused by the respective Tax alleles was determined by monitoring γH2AX levels over a 16-hr period post bleomycin treatment (see supplemental information S5 Fig for quantitation).

induction. Finally, when WT, M22 and M47 alleles were stably expressed in HeLa-G: ΔN-IκBα cells via lentiviral vectors (Fig 3E), WT and M47, but not M22 or Tax-negative control, repressed DDR as indicated by persistent and strong γH2AX accumulation post-bleomycin treatment (Fig 3F). These results support the notion that robust nuclear TSS formation via RNF8 activation causes sequestration of DDR mediators, leading to DDR repression.

## Loss of RNF8 dampens HTLV-1 viral replication

As Tax utilizes RNF8 to activate a large ensemble of Ser/Thr kinases and signaling pathways [15, 33], we investigated whether RNF8 might play a role in viral replication. To this end, wild-type and RNF8-null HeLa cells (HeLa-GΔ$^{RNF8}$) were transfected with LTR-Luc and an NF-κB

reporter, E-selectin promoter-Luc (E-sel-Luc), respectively in the presence or absence of Tax (CMV-Tax). As shown in Fig 4A & 4B, Tax-driven E-sel-Luc and LTR-Luc reporter activities became significantly reduced in the absence of RNF8. In addition, Tax expression as driven by the CMV promoter also became reduced in HeLa-GΔ$^{RNF8}$ (compare lane 3 and 4), likely due to the involvement of NF-κB in the activity of the CMV promoter. As such, whether the reduction in LTR trans-activation in RNF8-null cells is due to the loss of kinase signaling or a reduction in Tax expression is uncertain. To assess directly the impact of RNF8 depletion on viral gene expression, we used an ATL cell line, ATL-T, which is productively infected by HTLV-1 and chronically produces infectious viral particles. RNF8 expression in ATL-T was silenced by the stable transduction of lentiviral vectors that expresses RNF8-targeting shRNAs. As shown in Fig 4D, RNF8 silencing significantly reduced HTLV-1 capsid (p24) and matrix (p19) expression in ATL-T, demonstrating the importance of RNF8 activation in HTLV-1 replication.

## RNF8 down-regulation occurs frequently in ATL cells

As RNF8 is crucial for Tax-mediated IKK/NF-κB activation and viral replication, we examined its mRNA and protein expression in a collection of 7 established ATL (MT1, ATL-2, ATL-T, TLOm1, ED, ATL35T, and ATL55), 2 HTLV-1-transformed (C8166 and MT4), and 3 HTLV-1-negative (SupT1, Jurkat, and CEM) T-cell lines. As shown in Fig 5A & 5B, the levels of RNF8 protein and mRNA in 6/7 ATL cell lines (with the exception of ED) and 2/2 HTLV-1-transformed T cell lines were low or undetectable, but not so in HTLV-1-negative controls (SupT1, Jurkat, and CEM) (Fig 5A & 5B). This contrasts with the control glyceraldehyde 3-phosphate dehydrogenase (GAPDH) whose levels were similar across all cell lines analyzed (Fig 6A). We next asked if RNF8 expression is repressed in ATL cells of patients. To this end, the whole genome mRNA expression profiles of 52 ATL patients and 21 healthy volunteers (GSE33615 in the NCBI gene expression omnibus repository) [34] were analyzed. In agreement with Fig 5A & 5B, a significant reduction in RNF8, but not glucose 6-phosphate isomerase (GPI) or GAPDH mRNA expression was detected in the majority of ATL patients (Fig 5C upper left panel). Importantly, RNF8 down-regulation occurs across all ATL types (Fig 5C upper right panel), suggesting that it emerges early in the course of disease development. In agreement with the notion that RNF8 is specifically targeted during HTLV-1 infection, a similar analysis of diffuse large B cell lymphoma (DLBCL, n = 76) (GSE83632) revealed no difference in RNF8 expression compared to the healthy control (n = 87).

## Discussion

HTLV-1 viral trans-activator/oncoprotein Tax is a potent clastogen and the principal driver of the genomic instability in ATL, but the underlying mechanism(s) is (are) not fully understood. In this study, we have established a causal connection between DDR repression in HTLV-1-infected cells and the aberrant activation of the ubiquitin E3 ligase, RNF8, by Tax. Our data indicate that the previously described Tax-induced nuclear speckle structures (TSS) [13] co-localize with K63-linked polyubiquitin chains, and their formation is strictly dependent on RNF8 and its activation by Tax. In aggregate, our published results and current data indicate that the RNF8 hijacked by Tax assembles K63-linked polyubiquitin chains in both cytosolic and nuclear compartments. The cytosolic K63-linked polyubiquitin chains further enlist LUBAC to produce hybrid K63-M1 polyubiquitin chains for the recruitment and activation of TAK1 and IKK, leading to potent NF-κB activation [15, 22, 35]. The nuclear K63-linked polyubiquitin chains promote the formation of TSS, which, in essence, constitute pseudo DNA damage signaling scaffolds that mislocalize DDR factors, disrupt DDR signaling, and suppresses DSB repair. Our results further indicate that HTLV-1 viral replication is enhanced by

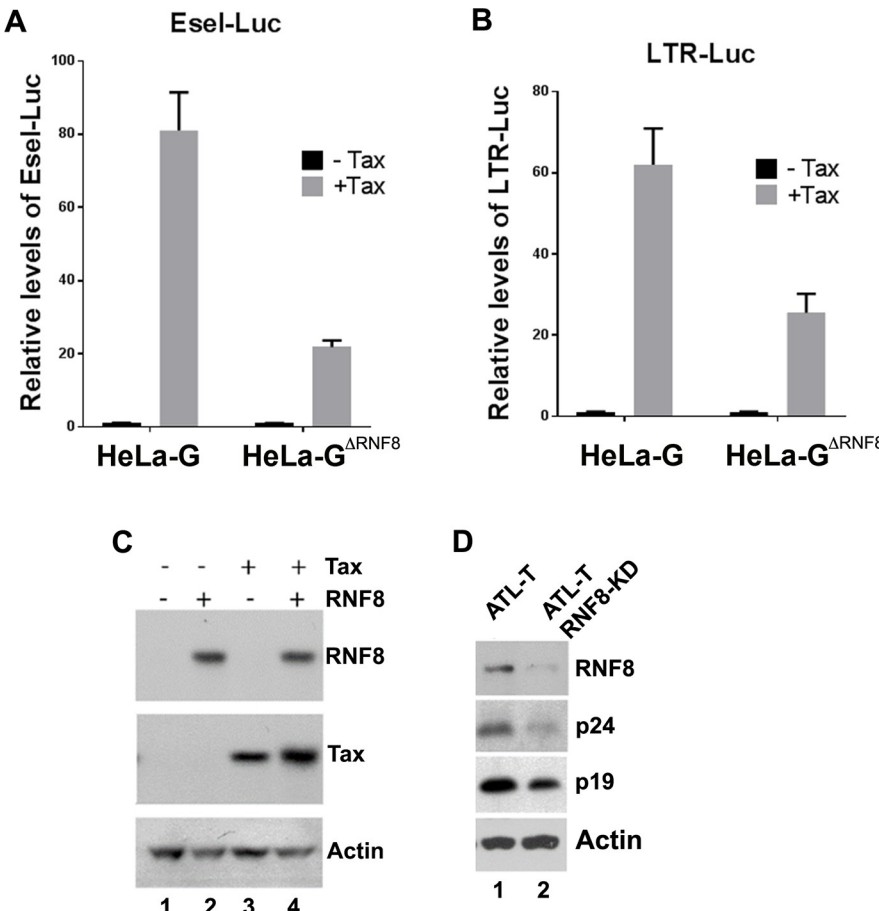

**Fig 4. Loss of RNF8 reduces Tax-mediated LTR and NF-κB activation, and viral protein expression. (A & B)** NF-κB luciferase reporter: E-selectin-Luc (Esel-Luc) or HTLV-1 LTR luciferase reporter: LTR-Luc was transfected into HeLa-G and HeLa-G$^{\Delta RNF8}$ cells in the presence or absence of Tax (50 ng) for 48 hrs. Luciferase activities and fold activation by Tax in RNF8-wildtype and RNF8-null cells were measured and calculated as previously described [15]. **(C)** Tax expression levels in transfected HeLa-G$^{\Delta RNF8}$ (lane 3) and HeLa-G (lane 4) cells were determined by immunoblotting. Un-transfected cells are included in lanes 1 and 2 as controls. **(D)** ATL-T and its progeny knocked down for RNF8 expression (ATL-T RNF8-KD) were immunoblotted for RNF8, HTLV-1 capsid and matrix proteins: p24 and p19, and β-actin.

RNF8. This likely occurs through the activation of TAK1 and downstream kinase signaling [33], which promotes viral and cellular gene expression. Finally, we show that RNF8 expression is down-regulated across ATL of all types. These results are summarized in a schematic diagram shown in Fig 6.

Present results and prior studies by others suggest that the genomic instability in ATL is caused by Tax through at least three mechanisms: (1) direct induction of DNA double-strand breaks; (2) sequestration of DDR mediators and repression of DNA damage repair via TSS formation; and (3) down-regulation of RNF8. The first two are driven by Tax and occur in HTLV-1-infected cells where Tax, a potent clastogen, acts as an initiator of carcinogenesis by destabilizing the genomes of infected cells [10–12]. As Tax is a major CTL target, its expression is often extinguished in ATL cells (>50%). The continuous proliferation of ATL cells is likely driven by the chronically expressed viral oncoprotein, HBZ, acting as a tumor promoter [36, 37].

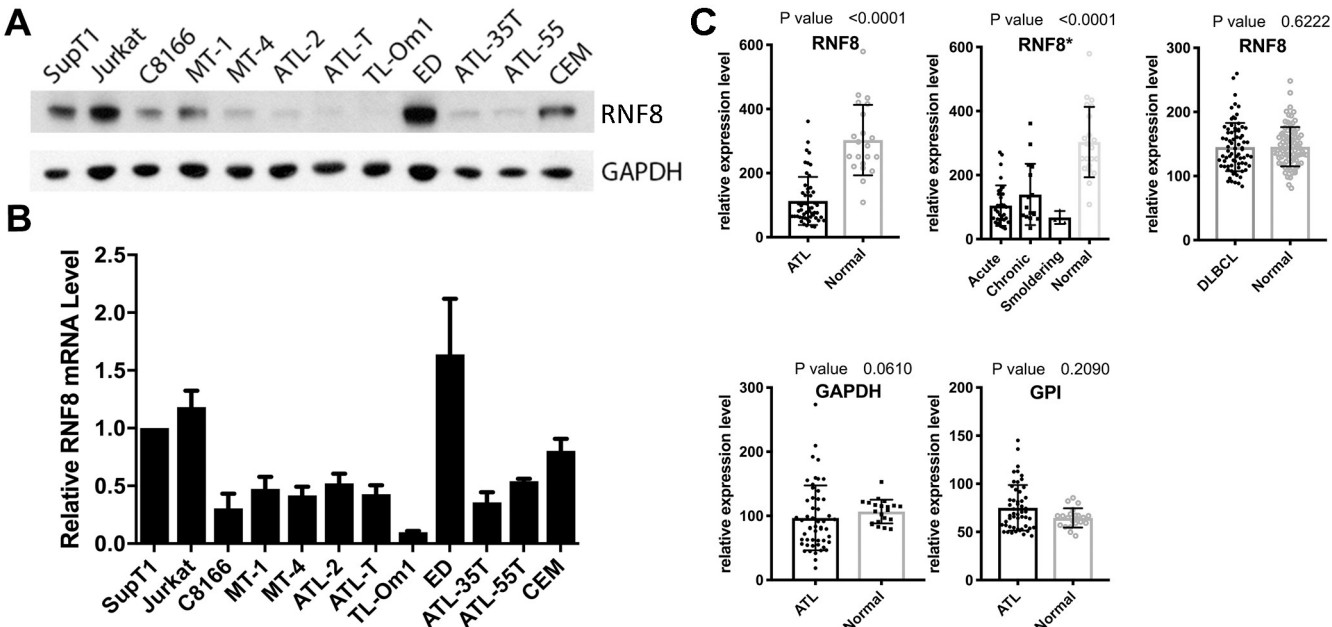

**Fig 5. Frequent down-regulation of RNF8 expression in ATL cells. (A)** Immunoblots of RNF8 and GAPDH in ATL, HTLV-1-transformed, and HTLV-1-negative T cell lines. ATL: MT-1, ATL-2, ATL-T, TL-Om1, ED, ATL-35T, & ATL-55; HTLV-1-transformed: C8166 & MT-4; and HTLV-1-negative: SupT1, Jurkat, & CEM. Antibodies used are as indicated, with GAPDH as a control for sample loading. **(B)** qRT-PCR analysis of RNF8 mRNA in ATL, HTLV-1-transformed, and HTLV-1-negative T cell lines. qRT-PCR of RNF8 and β-actin mRNAs was carried out for the cell lines listed in (A) and the ratio of the two calculated, and then normalized against that of the control SupT1 cells, and plotted. **(C)** RNF8 mRNA expression in the peripheral blood mononuclear cells (PBMCs) of ATL patients and CD4+ T cells of healthy volunteers. RNF8, glyceraldehyde-3-phosphate dehydrogenase (GAPDH) and glucose 6-phosphate isomerase (GPI) mRNA expression values of 52 ATL patients (ATL) and 21 healthy volunteers (Normal) were obtained from the NCBI ATL microarray data set (GSE33615) originally generated by Nakano et al [34, 42] using the CD4+ cells of healthy volunteers and the PBMCs of ATL patients (consisting of mostly ATL cells, Note: Most ATL cells contain a single copy of proviral DNA per cell. The PVLs of acute, chronic, and smoldering ATL were previously determined to be around 85.7, 76.4, and 23.6 per 100 PBMCs respectively [43]. Thus, ATL cells represent approximately 85.7%, 76.4%, and 23.6% of the PMBC populations in acute chronic, and smoldering ATL patients.) The data were derived using the Agilent-04850 whole human genome microarray platform. The expression values of RNF8, and GAPDH and GPI controls in each ATL patient (ATL) and healthy volunteer (Normal) were normalized to that of the cognate β-actin. Mann-Whitney test was performed to compare the normalized expression values using the GraphPad PRISM (8.0c) software. The RNF8 mRNA levels in all ATL patients vs healthy donors were shown in the upper left panel. The RNF8 expression data were further divided into acute, chronic and smoldering types and compared with those of healthy volunteers (Normal) using the Brown-Forsythe and Welch ANOVA tests (upper middle panel). As a control, a similar analysis of the relative RNF8 mRNA levels in the peripheral blood cells of 76 DLBCL patients versus 87 healthy donors (GSE83632) is shown in the upper right panel.

We have shown previously that RNF8 supports Tax-mediated IKK/NF-κB activation in a dose-dependent manner [15]. Because RNF8 down-regulation attenuates Tax-mediated cell signaling and HTLV-1 viral replication (Fig 4), we think it is positively selected during persistent viral infection to facilitate evasion from immune detection. Further, HTLV-1 infection and Tax expression are known to trigger a senescence response mediated by constitutively activated NF-κB. As the ablation of RNF8 abrogates canonical NF-κB activation by Tax [15], RNF8 down-regulation is expected to mitigate Tax senescence, contributing to the clonal expansion of HTLV-1-infected cells. In this vein, it is interesting to note that RNF8 deficiency occurs in ATL of all types (Fig 5C upper middle panel), suggesting that it likely emerges early during leukemia development.

In conclusion, both RNF8 dysregulation by Tax during viral infection and RNF8 down-regulation during chronic infection and progression to disease promote DDR impairment and genomic instability of ATL. DDR deficiencies impact cancer response to DNA-damaging radiation and chemotherapies. Whether RNF8 deficiency can be exploited for ATL treatment remains to be seen. Finally, whether the nuclear and cytosolic K63-polyubiquitin chains

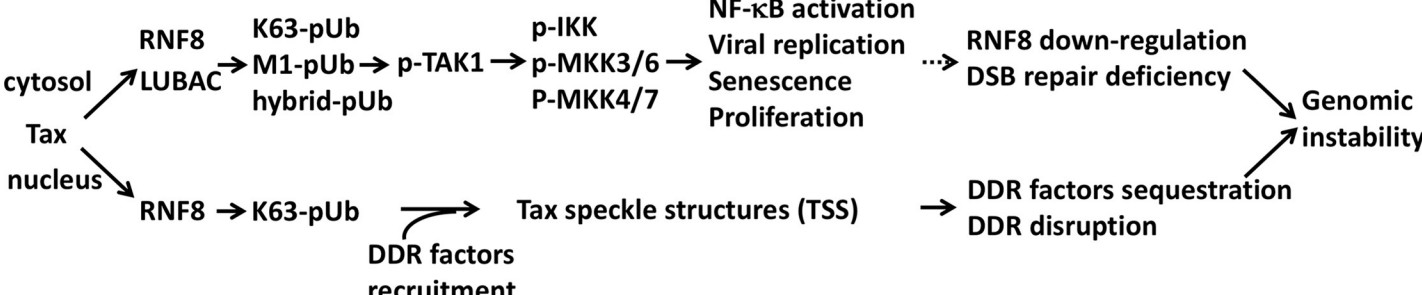

**Fig 6. A schematic summary of how RNF8 dysregulation and down-regulation impact the genomic instability of ATL cells.** (upper pathway) Aberrant activation of RNF8 by Tax leads to the accumulation of K63-pUb and K63-M1-hybrid pUb chains in the cytosol, and K63-pUb chains in the nucleus. The former triggers a cascade of kinase activation and phosphorylation, culminating in IKK/NF-κB activation, HTLV-1 replication, and senescence or proliferation depending on the cellular context. Cytotoxic T lymphocyte killing of virus-producing cells and senescence induction select for RNF8 down-regulation and silencing of viral gene expression. RNF8 down-regulation leads to DDR and double-strand break repair deficiency in ATL cells (lower pathway). Nuclear K63-pUb chains and Tax form microscopically visible Tax speckle structures that sequester DDR factors such as BRCA1, DNA-PK, and MDC1, disrupting DDR. RNF8 dysregulation by Tax and RNF8 down-regulation contribute to the genomic instability of HTLV-1-infected and ATL cells.

assembled by RNF8 during infection additionally disrupt other cellular processes such as cytokinesis [38–40] also awaits future investigation.

## Materials and methods

### Cell Lines and Cell Culture

Human T cell lines were cultured in RPMI-1640 media (Hyclone) supplemented with 10% fetal bovine serum (FBS), L-glutamine, 100U/ml penicillin and streptomycin and maintained in 5% $CO_2$ at 37˚C. HeLa-G and its progenies were grown in DMEM media (Quality Biologicals) containing 10% FBS. Derivation of the RNF8-null HeLa cell line, HeLa-GΔ$^{RNF8}$, has been previously reported [15]. The U2OS cell line ablated for RNF8 expression was similarly derived except a lentiviral vector that co-expresses an RNF8 guide RNA and Cas-9 together with a puromycin-resistance gene was used.

### Derivation of HTLV-1-infected HeLa-G: ΔN-IκBα clones

HTLV-1 infections were performed in a 10 cm dish by co-culturing HeLa-G: ΔN-IκBα cells (1-2x10^6) with HTLV-1-producing MT2 cells (3x10^6) that have been mitotically inactivated by mitomycin C treatment (10 μg/ml for 2 hrs). The co-culture was carried out in the presence of polybrene (8 μg/ml) for 24 hrs. MT2 cells were then removed by washing with phosphate buffered saline (PBS). Fresh media was then added and cells grown for an additional 24 hrs, and then harvested. GFP-positive cells were isolated using a cell sorter (BD FACSAria) housed in a lamella flow hood under aerosol-protection condition. Sorted cells were plated at low density on a 15-cm dish. After a week, individual colonies were picked and transferred into 96-well plates and further screened for the integrated HTLV-1 genome by PCR and immunoblotting.

### Immunoblotting

Cells were harvested and lysed in lysis buffer (Cell Signaling). Standard methods were used for immunoblotting. Samples containing 25 μg of whole cell lysates as determined by BCA assay (Promega) were loaded on 13.5% Tris-glycine gels. The HTLV-1 Tax mouse hybridoma monoclonal antibody 4C5 was generated in our laboratory as previously described [41]. All other antibodies were commercially available and as listed.

## Immunofluorescence

U2OS or HeLa cells were seeded onto 22-mm diameter coverslips in 6-well plates at $4x10^5$ cells/well. Twenty hours later, cells were transfected with a Tax-expressing plasmid for native Tax or Stag-Tax-GFP fusion using the Lipofectamine[TM] 3000 (ThermoFisher) transfection reagent according to the manufacturer's protocol for 48 hrs. Transfected cells were then washed twice with PBS, fixed in 4% paraformaldehyde and permeabilized with cold methanol. Where appropriate, coverslips were further incubated with a primary mouse monoclonal anti-Tax antibody or a primary mouse monoclonal anti-RNF8 antibody in 3% BSA in PBS at 4˚C overnight, followed by 2 washes in PBS+0.1% Tween-20, and 2 washes in PBS. A rhodamine- or fluorescein-conjugated secondary antibody and the fluorescein- or tetramethylrhodamine (TAMRA)-labeled K63 TUBE (LifeSensors) were then added and incubated for 1 hr at room temperature followed by 2 washes in 3% BSA in PBS and 2 washes in PBS. Coverslips were then mounted in Vectashield containing DAPI (Vector Laboratories, Burlingame, CA). Fluorescent images were acquired using a Zeiss LSM 510 confocal microscope at 63X magnification with a 2.0 x zoom, and photographed with the Image Browser software (Carl Zeiss, Jena, Germany). Immunofluorescence of HEK293 cells was carried out similarly except the DNA transfection was performed using the calcium phosphate precipitation method.

## RNA extraction and real-time quantitative PCR (RT qPCR)

Total mRNA from each T cell line was isolated using the PARIS kit (Ambion) according to the manufacturer's protocol. Contaminating genomic DNA was removed using the turbo DNA-free kit (Ambion). Complementary DNA (cDNA) was synthesized from 500 ng of RNA in a total volume of 10 μl using the iScript reverse transcription super mix (Biorad). For RNF8 and β-actin mRNA quantitation, RT-qPCR was performed using 2 μl of the cDNA as template in a 20 μl reaction and the BIO-RAD PrimePCR[TM] SYBR® Green Assays: RNF8, Human and ACTB, Human respectively. Relative mRNA levels were calculated using the $2^{-\Delta Ct}$ method [34]. The RNF8 mRNA level in each sample was normalized to that of the β-actin mRNA. The RNF8/ β-actin mRNA ratios of ATL and HTLV-1-transformed T cell lines were further normalized against that of the HTLV-1-negative CD4+ SupT1 T cell line, and plotted.

## S-tagged Pull-Down Assay

Pull-down of S-tagged wildtype, M47 and M22 Tax was performed as previously reported [15].

## Transfection and luciferase assays

Cells ($3X10^5$) were seeded into a 24-well plate overnight. After 16 hrs, DNA transfection was carried out using the Fugene HD reagent (Promega). For each reporter assay, 200 ng of each reporter plasmid, HTLV-1 LTR-Luc and E-selectin-Luc, respectively, together with 50 ng of a Tax expression plasmid, BC12-Tax, were used. All transfections were performed in triplicates. Twenty nanograms of a control luciferase plasmid pGL3-Luc (firefly) or pRL-TK (renilla) were also included in each transfection. The total DNA amount (500 ng) was kept constant by adding an empty vector plasmid, pcDNA3.1. After 48 hrs, cells were harvested, and luciferase activity measured using the Dual-Luciferase Reporter Assay System (Promega) according to the manufacturer's instructions. Transfection efficiencies were normalized using either TK-renilla or PGL3-Luc. Data are mean ± s.d. from at least three independent experiments.

### Lentivirus vectors preparation and transduction

Lentiviral vectors carrying shRNA or CRISPR-Cas9 cassettes that target RNF8 were prepared as previously reported [15]. HeLa-G and U2OS cells were transduced with the lentiviral vectors in DMEM supplemented with 10% fetal bovine serum and selected in the same medium containing puromycin (1 μg/ml). Cell clones silenced or deleted for the RNF8 gene were isolated after limiting dilution and confirmed by immunoblotting.

### Data quantitation and statistical analysis

The accession number for the ATL microarray data set is GSE33615. PRISM 8.0 and Image J were used respectively for the statistical analysis of gene expression values and quantitation of immunoblots. The expression value of RNF8 in each individual was normalized against that of the cognate β-actin gene. The relative expression values of ATL patients and healthy volunteers were compared using the Mann-Whitney test or Kruskal-Wallis test provided in PRISM 8.0.

### Disclaimer statement

The opinions and assertions expressed herein are those of the author(s) and do not necessarily reflect the official policy or position of the Uniformed Services University or the Department of Defense.

### Supporting information

**S1 Fig. Quantitation of the γH2AX levels in Fig 1A.** γH2AX levels of HeLa-G: ΔN-IκBα and its progeny HeLa-G: ΔN-IκBα:HTLV-1 cells in Fig 1A were quantified using Image J and normalized to the β-actin (Actin) loading control. The values were presented relative to the untreated HeLa-G: ΔN-IκBα.
(TIF)

**S2 Fig. Quantitation of the γH2AX levels in Fig 1B.** γH2AX levels of HeLa-G and HeLa-G^(ΔRNF8) cells in Fig 1B were quantified and normalized in S1. The values were presented relative to the untreated wild-type HeLa-G control.
(TIF)

**S3 Fig. Quantitation of the γH2AX levels in Fig 1C.** γH2AX levels of HeLa-G^(ΔRNF8) cells transduced with either Ad-GFP (Control) or Ad-RNF8 in Fig 1C were quantified and normalized as above and presented relative to the untreated HeLa-G^(ΔRNF8) cells transduced with Ad-GFP.
(TIF)

**S4 Fig. Quantitation of the polyubiquitin levels in Fig 3A.** The total polyubiquitin chain signals in Fig 3A were quantified using Image J and normalized to the β-actin (Actin) loading control of each experiment, and then to the normalized value of the untransfected control cells (denoted as "-").
(TIF)

**S5 Fig. Quantitation of the γH2AX levels in Fig 3F.** γH2AX levels of HeLa-G: ΔN-IκBα and its progenies expressing WT, M47 and M22 Tax were quantified using Image J, normalized to the GAPDH loading control and then to the untreated sample of HeLa-G: ΔN-IκBα cells.
(TIF)

## Acknowledgments

We thank Masao Matsuoka for the ATL cell lines and Cara Olsen for advice on statistical analysis.

## Author Contributions

**Conceptualization:** Huijun Zhi, Yik-Khuan Ho, Oliver John Semmes, Chou-Zen Giam.

**Data curation:** Huijun Zhi, Yik-Khuan Ho, Oliver John Semmes, Chou-Zen Giam.

**Formal analysis:** Huijun Zhi, Xin Guo, Yik-Khuan Ho, Nagesh Pasupala, Hampus Alexander Anders Engstrom, Oliver John Semmes, Chou-Zen Giam.

**Funding acquisition:** Chou-Zen Giam.

**Investigation:** Huijun Zhi, Oliver John Semmes, Chou-Zen Giam.

**Methodology:** Huijun Zhi, Xin Guo, Yik-Khuan Ho, Nagesh Pasupala, Hampus Alexander Anders Engstrom, Oliver John Semmes, Chou-Zen Giam.

**Project administration:** Oliver John Semmes, Chou-Zen Giam.

**Resources:** Oliver John Semmes, Chou-Zen Giam.

**Supervision:** Oliver John Semmes, Chou-Zen Giam.

**Validation:** Huijun Zhi, Oliver John Semmes, Chou-Zen Giam.

**Visualization:** Huijun Zhi, Xin Guo, Oliver John Semmes, Chou-Zen Giam.

**Writing – original draft:** Chou-Zen Giam.

**Writing – review & editing:** Huijun Zhi, Xin Guo, Oliver John Semmes, Chou-Zen Giam.

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
