## [Decision Letter · Decision Letter 0]

18 Feb 2020

Dear Prof. Giam,

Thank you very much for submitting your manuscript "RNF8 Dysregulation and Down-regulation During HTLV-1 Infection Promote Genomic Instability in Adult T-Cell Leukemia" for consideration at PLOS Pathogens. As with all papers reviewed by the journal, your manuscript was reviewed by members of the editorial board and by several independent reviewers. In light of the reviews (below this email), we would like to invite the resubmission of a significantly-revised version that takes into account the reviewers' comments.

The three expert reviewers agree that your manuscript reports some original and potentially important observations, but that some clarification and further experiments are required. 

We cannot make any decision about publication until we have seen the revised manuscript and your response to the reviewers' comments. Your revised manuscript is also likely to be sent to reviewers for further evaluation.

Sincerely,

Charles R. M. Bangham

Associate Editor

PLOS Pathogens

Susan Ross

Section Editor

PLOS Pathogens

Kasturi Haldar

Editor-in-Chief

PLOS Pathogens

orcid.org/0000-0001-5065-158X

Michael Malim

Editor-in-Chief

PLOS Pathogens

orcid.org/0000-0002-7699-2064

The three expert reviewers agree that your manuscript reports some original and potentially important observations, but that some clarification and further experiments are required.

Reviewer's Responses to Questions

**Part I - Summary**

Reviewer #1: In this manuscript, “RNF8 dysregulation and downregulation during HTLV-1 infection promote genomic instability in Adult T-cell Leukemia”, Zhi et al. report new roles for ring finger protein 8 (RNF8) in HTLV-1 infection. First, they found that RNF8, a lysine 63 (K63)-specific ubiquitin E3 ligase, is involved in forming Tax-speckled structures (TSS) in the nucleus. They propose that Tax and K63 polyubiquitin chains assembled by RNF8 recruit and sequester DNA damage response factors in the TSS, leading to a deficiency in the ability of HTLV-1-infected cells to respond to DNA damage. Second, they determined that RNF8 expression is associated with an increase in HTLV-1 transcription, possibly through a signaling cascade initiated by RNF8-mediated activation of TGFβ-activated kinase 1 (TAK1) in the cytoplasm. Finally, they observed that RNF8 levels are reduced in ATL transformed cells. Therefore, according to the effect of RNF8 on HTLV-1 transcription, the selection of HTLV-1-infected cells that escape elimination by the immune system might involve down-regulated RNF8 expression. Overall, this an interesting study that contributes to our knowledge of the multiple roles of RNF8 in HTLV-1 infection. The following points will complement some of the data presented.

Reviewer #2: HTLV-1 is a human retrovirus which induces adult T-cell leukemia-lymphoma (ATLL), and a viral protein Tax is thought to be important for the oncogenic mechanisms. The authors of this manuscript previously reported that Tax interacts with a host ubiquitin E3 ligase, RNF8, and activates the NFkB signaling, which is a critical pathway for development of ATLL. In this paper, Zhi et al. show the novel aspects of RNF8 in HTLV-1-infected and Tax-expressing cells: 1) nuclear Tax speckle structures (TSS), which are associated with DDR impairment, is dependent on RNF8, 2) RNF8 is also involved in viral replication, and 3) RNF8 is down-regulated in ATLL cell lines and primary ATLL cells. It is clearly shown that RNF8 is critical for TSS formation by using CRISPR/CAS9 technique; however, it remains unclear how RNF8 induces TSS formation and inhibits DDR. This study is interesting, but there are several concerns to be addressed.

Reviewer #3: In the present study, Zhi et al. provide evidence suggesting that genomic instability in adult T-cell leukemia, a key hallmark of ATL, might be due to the ability of Tax to aberrantly activate the RNF8 K63-specific ubiquitin E3 ligase (critical for DNA double-strand break repair signaling). In the cytoplasm this activation would trigger the canonical IKK:NF-kB pathway, viral replication, RNF8 downregulation and defective DSB repair. In the nucleus K63-pUb sequesters DDR factors into Tax-speckle structures, also leading to genomic instability.

The topic of the study is interesting and represents a logical extension of previous studies from the group.

However some of the data do not convincingly support the conclusion of the authors and limit the scope of this work.

**Part II – Major Issues: Key Experiments Required for Acceptance**

Reviewer #1: 1) In Figure 1A, quantification of several experiments would be helpful to visualize the DDR impairment induced by HTLV-1.

2) Why do the authors switch between HeLa-G delta-RNF8 and U2OS delta-RNF8 cells in Figure 2? Since the DNA damage experiments are shown in HeLa cells, it will make more sense to show colocalization of Tax and K63-pUb in the same cell lines.

3) The same question regarding Figure 2 also applies to Figure 3: will the TSS be similar in HeLa-G:deltaN-IKBalpha cells?

4) The authors should determine whether Tax M47 and M22 interact with RNF8.

5) The new role of RNF8 in mediating activation of HTLV-1 transcription is interesting and could be further expanded. Will ectopic expression of RNF8 rescue the downregulation of LTR-Luc transcription observed in RNF8KD cells? The authors propose that TAK1 and downstream effects could be responsible for activation of the LTR promoter. Could they use a TAK1 inhibitor to support their claim? Tax levels could also be shown in the western blot in Figure 4D.

Reviewer #2: 1. The authors show that RNF8 is required for appropriate DDR after genotoxic stress (Fig. 1B and C), and HTLV-1 infection impaired DDR even in the presence of RNF8 (Fig. 1A and D). They also think that Tax induced activation of RNF8, but DDR was impaired. These findings suggest that Tax suppresses the function of RNF8; however, there is little description about the mechanisms how activation of RNF8 by Tax impairs DDR. It is known that RNF8 interacts with MDC1, which is one of the components of Tax speckle structures (TSS). Does TSS contain RNF8 in HTLV-1-infected or Tax-overexpressing cells? If so, does Tax inhibit the interaction of RNF8 with other proteins in TSS, or its dissociation from the complex? It would be better if the authors could show some experimental data to suggest the mechanisms.

2. In Fig. 3, the authors show that TaxM22 mutant could not induce TSS and impair DDR, indicating the NFkB pathway is involved in this machinery. The authors should discuss about the possible mechanisms for TSS formation by Tax-mediated NFkB activation.

3. It is evident that RNF8 is down-regulated in ATLL cell lines and primary ATLL cells. The author reported that RNF8 is important for NFkB activation by Tax. Is there any correlation between expression level of RNF8 and activation of NFkB in those cells?

Reviewer #3: Major criticisms

• Fig. 1: The statement “Notably, in the absence of bleomycin treatment, a low level of gH2AX was also observed in HeLa-G:DN133 IkBa:HTLV-1, but not in uninfected HeLa-G:DN-IkBa cells (compare Fig. 1A lanes 2 & 1)” (lines 132-133). This is not at all evident in the blot shown. Also the differences in panels B and C appear weak and unconvincing.

• Especially given these marginal differences, all immunoblots should be run in repeats, signals should be carefully quantitated and means and error bars should be shown. The number of repeats should be sufficient to support meaningful statistical analyses.

• Fig. 3 should also show RNF8and γH2AX expression and localization.

• Most of the work presented is carried out in cell lines and only panel 5C shows expression data (from a published microarray dataset) in samples from ATL patients. It would be interesting to have data on the percentage of ATL cells in these PBMC samples and on the levels of expression of Tax and other viral genes in these samples.

**Part III – Minor Issues: Editorial and Data Presentation Modifications**

Reviewer #1: 1) Lane 177: "lanes 13" should be corrected.

2) Lane 214: pUbs assembly does not look reduced in Tax M47 compared to Tax wt (Fig. 3A). Quantification might help visualize the difference.

3) Lane 220, Fig. 4D should be 3E.

Reviewer #2: 1. Fig.1 A-D: Differences and dynamics of the expression of gH2AX are not clear. Intensities of all bands should be quantified and normalized by an imaging software, and the dynamics of their expression levels should be shown as the graphs in the different panels of Figure 1.

2. Page 8, line 177: There is a typo in the terms “(lanes 1 & 2, also see Fig. 1A & B lanes 13 for comparison)”. There are no lanes “13” in Fig. 1A & B.

3. Page 10, line 220: “(Fig. 4D)” should be “(Fig. 3E)”. In addition, there is no explanation about Fig. 3D in the main text.

Reviewer #3: (No Response)

PLOS authors have the option to publish the peer review history of their article (what does this mean?). If published, this will include your full peer review and any attached files.

Reviewer #1: No

Reviewer #2: No

Reviewer #3: No
---

## [Editor Report · Decision Letter 1]

11 May 2020

Dear Prof. Giam,

We are pleased to inform you that your manuscript 'RNF8 Dysregulation and Down-regulation During HTLV-1 Infection Promote Genomic Instability in Adult T-Cell Leukemia' has been provisionally accepted for publication in PLOS Pathogens.

Best regards,

Charles R. M. Bangham

Associate Editor

PLOS Pathogens

Susan Ross

Section Editor

PLOS Pathogens

Kasturi Haldar

Editor-in-Chief

PLOS Pathogens

orcid.org/0000-0001-5065-158X

Michael Malim

Editor-in-Chief

PLOS Pathogens

orcid.org/0000-0002-7699-2064

Thank you for the comprehensive and reasoned response to the comments and questions raised by the reviewers.
---

## [Editor Report · Acceptance letter]

18 May 2020

Dear Prof. Giam,

We are delighted to inform you that your manuscript, "RNF8 Dysregulation and Down-regulation During HTLV-1 Infection Promote Genomic Instability in Adult T-Cell Leukemia," has been formally accepted for publication in PLOS Pathogens.

Best regards,

Kasturi Haldar

Editor-in-Chief

PLOS Pathogens

orcid.org/0000-0001-5065-158X

Michael Malim

Editor-in-Chief

PLOS Pathogens

orcid.org/0000-0002-7699-2064